# Chronic Voluntary Morphine Intake Is Associated with Changes in Brain Structures Involved in Drug Dependence in a Rat Model of Polydrug Use

**DOI:** 10.3390/ijms242317081

**Published:** 2023-12-03

**Authors:** María Elena Quintanilla, Paola Morales, Daniela Santapau, Alba Ávila, Carolina Ponce, Pablo Berrios-Cárcamo, Belén Olivares, Javiera Gallardo, Marcelo Ezquer, Mario Herrera-Marschitz, Yedy Israel, Fernando Ezquer

**Affiliations:** 1Molecular and Clinical Pharmacology Program, Institute of Biomedical Science, Faculty of Medicine, Universidad de Chile, Santiago 7610658, Chile; equintanilla@uchile.cl (M.E.Q.); pmorales@uchile.cl (P.M.); mh_marschitz@uchile.cl (M.H.-M.); yisrael@uchile.cl (Y.I.); 2Department of Neuroscience, Faculty of Medicine, Universidad de Chile, Santiago 7610658, Chile; 3Research Center for the Development of Novel Therapeutic Alternatives for Alcohol Use Disorders, Santiago 7610658, Chile; 4Center for Regenerative Medicine, Faculty of Medicine Clínica Alemana-Universidad del Desarrollo, Santiago 7610658, Chile; dsantapau@udd.cl (D.S.); aavilas@udd.cl (A.Á.); pablo.berrios@udd.cl (P.B.-C.); javiera.gallardo@udd.cl (J.G.); mezquer@udd.cl (M.E.); 5Center for Medical Chemistry, Faculty of Medicine Clínica Alemana-Universidad del Desarrollo, Santiago 7610658, Chile; molivares@udd.cl

**Keywords:** animal model, neurodegeneration, opioid addiction, brain damage, neuroinflammation, polydrug, dependence

## Abstract

Chronic opioid intake leads to several brain changes involved in the development of dependence, whereby an early hedonistic effect (liking) extends to the need to self-administer the drug (wanting), the latter being mostly a prefrontal–striatal function. The development of animal models for voluntary oral opioid intake represents an important tool for identifying the cellular and molecular alterations induced by chronic opioid use. Studies mainly in humans have shown that polydrug use and drug dependence are shared across various substances. We hypothesize that an animal bred for its alcohol preference would develop opioid dependence and further that this would be associated with the overt cortical abnormalities clinically described for opioid addicts. We show that Wistar-derived outbred UChB rats selected for their high alcohol preference additionally develop: (i) a preference for oral ingestion of morphine over water, resulting in morphine intake of 15 mg/kg/day; (ii) marked opioid dependence, as evidenced by the generation of strong withdrawal signs upon naloxone administration; (iii) prefrontal cortex alterations known to be associated with the loss of control over drug intake, namely, demyelination, axonal degeneration, and a reduction in glutamate transporter GLT-1 levels; and (iv) glial striatal neuroinflammation and brain oxidative stress, as previously reported for chronic alcohol and chronic nicotine use. These findings underline the relevance of polydrug animal models and their potential in the study of the wide spectrum of brain alterations induced by chronic morphine intake. This study should be valuable for future evaluations of therapeutic approaches for this devastating condition.

## 1. Introduction

Opioid use disorder (OUD) is a worldwide public health emergency, with over 17 million opioid dependent individuals [1]. It is a devastating neuropsychiatric condition that leads to severe suffering for patients and their families. In recent years, opioid overdose deaths, mainly caused by respiratory depression, have continued to rise, resulting in over 130,000 deaths worldwide in the last year (WHO 2021). Additionally, chronic opioid intake is frequently associated with several brain alterations [2], greatly affecting the quality of life of patients. 

It has been consistently reported that the oral route is the preferred route of administration for prescription opioids, including oxycodone, hydrocodone, and morphine, and over 90% of abusers of prescription opioids report their oral ingestion for non-medical purposes [3,4]. Thus, the generation of animal models of voluntary opioid oral intake represents an important tool for identifying cellular and molecular brain alterations induced by chronic opioid use.

In recent years, it has become clear that drug dependence is shared among many drugs, and polydrug use is the norm rather than an exception [5]. Importantly, polydrug use is particularly prevalent among patients with opioid use disorder [6]. In the Netherlands, half of methadone users reported cocaine as their secondary drug. In a study in the United States, 60% of patients in a methadone maintenance program used crack cocaine and alcohol [7]. In Barcelona, more than one-third of heroin-addicted patients were found to be polydrug users when they began methadone maintenance treatment [8]. 

Animal models have been valuable for studying various aspects of drug addiction [9], and the above studies suggest that animals selectively bred to prefer one drug may show preference for other drug types as well. Thus, studies of chronic drug use in animals bred for one drug preference could be closer to natural models for different addictive substances. The Wistar-derived outbred UChB rat has for decades been selected for its high alcohol preference [10]. Subsequently, it was demonstrated that the UChB rats chronically self-administer nicotine [11], showing that many of the mechanisms of nicotine self-administration and relapse are shared with those induced by ethanol [11,12,13,14,15]. Thus, it has been postulated that UChB rats will develop dependence on morphine as well, exhibiting the cortical and striatal alterations reported for chronic alcohol and nicotine intake [11,12,13] as well as those documented for opioids [16], including neurological impairments such as deficits in cognition and executive function reflective of opioid-induced neuronal toxicity [2,17,18]. 

Opioid-induced alterations have mainly been associated with the direct induction of apoptotic pathways in neurons [19,20] as well as with opioid-induced generation of neuroinflammation and oxidative stress in specific brain areas, including the prefrontal cortex, neostriatum, and nucleus accumbens [19,21,22,23], which are brain areas mainly associated with drug dependence and loss of control over drug intake [24]. In the microglia and astrocytes, opioids, like other addictive drugs, directly activate the foreign molecule sensing toll-like receptor 4 (TLR-4), leading to the activation of the transcription factor NF-κB and production of pro-inflammatory mediators such as TNF-α, IL-1β, and IL-6 [25,26]. Additionally, morphine exposure significantly increases the production of the CC-chemokine ligand 5 (CCL5), leading to pro-inflammatory glial activation [27]. The increased production of these pro-inflammatory molecules induces neuronal death by initiating programmed cell death pathways [28]. Furthermore, as with many drugs of abuse, increases in extracellular glutamate levels generated by opioid-induced down-regulation of the main glutamate transporter in astrocytes, GLT-1, have been associated with excitotoxic processes and neuronal death [29,30]. 

Oligodendrocytes, the myelin-producing cells of the central nervous system, are highly affected by chronic opioid exposure, as opioids can directly induce apoptosis of myelinating oligodendrocytes after activation of opioid receptors, leading to an important reduction in myelination [31]. Similar effects have been reported for chronic alcohol exposure [32]. Moreover, opioid-mediated production of pro-inflammatory cytokines by microglia and astrocytes can impair the differentiation of oligodendrocyte precursor cells and increase the apoptosis of mature oligodendrocytes, ultimately resulting in additional demyelination [33,34] and axonal degeneration [35] that contribute to the impairment of cognitive abilities commonly observed in opioid-dependent individuals.

In recent studies, we showed that following one-week intraperitoneal morphine administration, UChB rats voluntarily consumed escalating amounts of morphine until reaching a steady state [36]. Rats typically avoid consuming morphine solutions due to their bitter taste. Therefore, intraperitoneal morphine administration enables the induction of a level of dependence, subsequently promoting voluntary morphine consumption [36].

From the above, the aim of the studies in the present communication was to determine whether the oral voluntary morphine intake in this rat model (a) leads to morphine dependence, as evidenced by a naloxone-precipitated morphine withdrawal reaction; (b) induces prefrontal cortex alterations known to be associated with the loss of control over drug intake, including demyelination, axonal degeneration, and reduction in glutamate transporter levels; and (c) leads to the development of neuroinflammation and oxidative stress, as has been shown following alcohol and nicotine self-administration. These parameters were evaluated after 14 weeks of voluntary morphine intake.

## 2. Results

### 2.1. Generation and Expression of Morphine Dependence: Chronic Voluntary Morphine Intake Induces Severe Withdrawal Syndrome in UChB Rats upon Naloxone Administration

To evaluate whether chronic voluntary morphine intake could induce dependence in rats selectively bred for high ethanol consumption [10,37], UChB rats were pretreated with daily intraperitoneal administration of morphine hydrochloride (40 mg/kg) for 9 days. This pretreatment aimed to promote subsequent voluntary morphine intake of a bitter-tasting morphine solution otherwise rejected by animals. After discontinuation of morphine injections, the animals were provided free access to a two-bottle choice between water or a morphine sulphate solution, allowing the animals to control their drug intake. The concentration of the morphine solution was increased (ranging from 6–90 mg/L) on successive days (day 1 to 40) as previously described [36]. On day 41, the animals were offered a three-bottle choice between water, morphine sulphate 80 mg/L, and morphine sulphate 90 mg/L (Appendix A). On day 50, when the animals had reached a stable morphine consumption (15.6 ± 0.8 mg/kg/day), morphine dependence was assessed by the development of a withdrawal syndrome by intraperitoneal administration of the opioid antagonist naloxone (5 mg/Kg). 

Upon naloxone administration, the morphine-drinking animals exhibited a strong withdrawal syndrome characterized by significant increases in various withdrawal signs during a 30 min observation period, including weight loss (Figure 1A), diarrhea (Figure 1B), forepaw tremors (Figure 1C), chewing (Figure 1D), jumping (Figure 1E), climbing (Figure 1F), rearing events (Figure 1G), and total withdrawal score (Figure 1H) compared to only water-drinking animals. These results indicated that 50 days of voluntary morphine intake induced substantial morphine dependence in UChB rats. 

As expected, naloxone administration led to a significant reduction in voluntary morphine intake upon morphine reassess, which gradually returned to pre-naloxone levels five days later (Appendix A). After confirming morphine dependence on day 50, the animals were allowed to continue to voluntarily drinking morphine solutions up to day 96, reaching an intake of 14.64 ± 0.52 mg morphine/kg/day (mean ± SEM) (Figure 2A) and a morphine preference of 83% versus water (Figure 2B). Chronic morphine intake was not associated with alterations in body weight compared to animals only exposed to water (Appendix A).

### 2.2. Morphine Dependence Is Associated with Alterations in Prefrontal Cortex

Chronic opioid administration has previously been associated with severe structural brain alterations, including demyelination and neurodegeneration [35]. However, it remains unclear whether these opioid-induced brain alterations are present in animals which display dependence. To investigate the above, animals were euthanized after 96 days of voluntary morphine intake and various structural alterations were assessed in the prefrontal cortex, a brain region highly affected by chronic morphine exposure [38]. We observed that chronic morphine intake led to a strong reduction in myelin basic protein (MBP) (Figure 3A) and myelin proteolipid protein (PLP) (Figure 3B) levels, which are considered specific markers of myelinating oligodendrocytes compared to water-drinking animals. The reduction of myelination was associated with a significant decrease in neurofilament-L levels, considered to be indicative of axonal damage [39] (Figure 3C).

The reductions of MBP and neurofilament-L protein levels in the prefrontal cortex of morphine-drinking animals, which were confirmed by immunofluorescence (Figure 4A,B), indicate that morphine dependence coexists with substantial structural alterations in the prefrontal cortex.

### 2.3. Morphine Dependence Is Associated with Functional Alterations in the Prefrontal Cortex

Reductions in the levels of GLT-1 and xCT, the glial glutamate transporters that regulate glutamate uptake, as well as of the dopamine transporter DAT, have been implicated in the development of morphine dependence [40]. Thus, we assessed the protein levels of these molecules in the prefrontal cortex by Western blot. The data indicated that 96 days of chronic morphine intake significantly reduced the levels of the glutamate transporter GLT-1 (Figure 5A) (*p* < 0.05) compared to only water-drinking animals without significantly affecting xCT (Figure 5B) or DAT levels (Figure 5C).

### 2.4. Morphine Dependence Is Associated with Neuroinflammation and Brain Oxidative Stress in: The Nucleus Accumbens Reinforcing Area

Glial cells play a central role in inducing and maintaining brain inflammation [41]. Thus, to investigate whether rats voluntarily consuming morphine for 96 days displayed neuroinflammation, astrocyte and microglial density were evaluated by immunofluorescence in the nucleus accumbens, a crucial brain region that controls opioid dependence and reward [42]. Data showed that morphine dependence after 96 days of voluntary morphine intake significantly increased astrocyte density (Figure 6A,C) (*p* < 0.001) and microglial density (Figure 6B,D) (*p* < 0.01) in this brain region compared to water-drinking animals. To correlate the increase in gliosis induced by voluntary morphine intake with other markers of neuroinflammation, the mRNA levels of several pro-inflammatory molecules were measured by RT-qPCR in the same brain region. Chronic morphine intake induced a significant increase in the mRNA levels of the pro-inflammatory chemokine CCL-5 in the nucleus accumbens (Figure 7A) (*p* < 0.05); this cytokine is released by astrocytes upon opioid receptor activation [43]. However, chronic morphine intake did not alter the mRNA levels of other pro-inflammatory cytokines, including IL1β (Figure 7B), TNF-α (Figure 7C), and IL6 (Figure 7D), compared to water-drinking animals.

Oxidative stress was assessed by determining the level of the lipid peroxidation marker malondialdehyde (MDA), a sensitive marker of brain cellular redox state [44], in the neostriatum, a critical brain structure of the reward system. As anticipated, chronic morphine intake led to a significant increase (*p* < 0.05) in MDA levels compared to water-drinking animals (Figure 8).

## 3. Discussion

Studies show that a rat bred to prefer ingesting ethanol of nicotine solutions over water can develop preference for an opioid as well. Chronic opioid intake led to (i) marked opioid dependence, as evidenced by the generation of strong withdrawal signs upon naloxone administration; (ii) prefrontal cortex alterations associated with the loss of control over drug intake, including significant demyelination, axonal degeneration, and reduction in levels of the glutamate transporter GLT-1; and (iii) striatal glial neuroinflammation and oxidative stress.

Noteworthily, the reverse may not hold true. It has been reported that rats bred to prefer an opioid do not show a preference for ethanol or cocaine [45], suggesting a directional susceptibility. However, the present study shows chronic morphine intake leading to many of the alterations reported in alcohol-consuming animals, including striatal neuroinflammation, oxidative stress [12], and demyelination [46].

Craving, which is an abnormal executive prefrontal function and loss of control, has been proposed to depend on prefrontal cortex dysregulation [24], likely influenced by prefrontal demyelination and reduction in levels of the glutamate transporter GLT-1, as reported here. Prefrontal cortex regulation influences the neuronal activity of the ventral tegmental area, feeding into the nucleus accumbens [47]. These authors have postulated that midbrain dopaminergic neurons integrate reward expectancy-related information from the prefrontal cortex.

A dysfunctional hypoactive prefrontal cortex executive function has been suggested to contribute to the loss of control over alcohol intake [48]. Furthermore, in heroin-dependent individuals, impulsive behavior is correlated with a reduction in prefrontal cortex gray matter volume [49]. Overall, the addictive process is accompanied by prefrontal cortex dysfunction [50]. While we have not studied a possible reversal of demyelination, studies in heroin users have indicated that abnormal cognitive control can be reversed [51,52], opening up a window of opportunity for the generation of new therapeutic treatments. 

In recent years, strong concerns regarding the effect of chronic opioid exposure on neural degeneration have emerged. Toxic effects of opioids, including demyelination and axon degeneration in various brain regions responsible for impulse control, reward, and motivation, has been described in a growing number of studies of heroin, methadone, oxycodone, and morphine-dependent patients [53,54,55,56]. Data suggest that opioid-induced demyelination leading to neuronal degeneration is a frequent finding [57]. In the present animal model, we observed that 14 weeks of voluntary morphine intake induced a severe reduction in MBP, PLP, and neurofilament-L protein levels in the prefrontal cortex. MBP and PLP synthesis by mature oligodendrocytes provides insulation to axons, protecting them from the damaging microenvironment, while neurofilament-L is a neuronal cytoplasmic protein highly expressed in large-caliber myelinated axons [39]. Thus, the reduction of these proteins suggests that prolonged morphine exposure leads to demyelination and axonal damage in the cortex and probably other brain regions. Our observations are in agreement with another study showing similar levels of reduction of MBP and neurofilament-L in the same brain region in an animal model of chronic oral oxycodone administration [35], a similar reduction of neurofilament-L levels in the prefrontal cortex of morphine-injected animals [58], and in the brain of opioid addicts [59], indicating that demyelination and axonal degeneration are common processes induced by different prescription opioids. 

Several preclinical and clinical studies have shown that neuroinflammation and brain oxidative stress are generated after chronic administration of most drugs of abuse, including morphine, and are strongly associated with opioid-induced neuronal damage, demyelination, white matter loss, and the perpetuation of drug intake and relapse [60,61]. Morphine consumption is shown to increase neuroinflammation by the activation of TLR4 in microglia and astrocytes, leading to the initiation of a signaling cascade that results in microglia and astrocyte activation and the production of several pro-inflammatory mediators [62,63]. Morphine intake promotes a rise in brain oxidative stress through a direct increase in the production of reactive oxygen species and NO synthesis after μ-opioid receptor activation [64] as well as by reduction of the antioxidant enzymatic machinery [60]. Increases in ROS and pro-inflammatory molecule levels result in neuronal and oligodendrocyte death and behavioral changes [28], including impaired cognitive functions. These molecular alterations characterize opioid tolerance and may represent the engines of relapse [60], as the administration of antioxidant and anti-inflammatory molecules has been shown to reduce voluntary morphine intake [36], morphine withdrawal [23], and morphine conditioned place preference [65].

It has been reported that morphine-mediated increases in microglia and astrocyte activation and the concomitant increase in the production of pro-inflammatory cytokines appear to be more pronounced in brain areas that participate in neuronal networks involved in controlling opioid abuse and reward, including the nucleus accumbens and the neostriatum [43,66]. Indeed, it has been suggested that activation of microglia in the nucleus accumbens is necessary for the reinstatement of morphine-seeking behavior [43]. In accordance with these affirmations, in the present animal model of oral morphine intake, 96 days of voluntary morphine consumption led to a significant increase in morphine-induced neuroinflammation as evidenced by an increase in astrocyte and microglia density in nucleus accumbens and a significant increase in MDA levels in the neostriatum, which is considered a reliable marker of oxidative stress [44]. Unexpectedly, we did not observe changes in mRNA levels of the classical pro-inflammatory cytokines IL1β, TNF-α, or IL6 in nucleus accumbens when comparing both experimental groups. However, mRNA levels of the chemokine CCL-5 were significantly increased in morphine-drinking animals. It has been reported that CCL-5 is produced by astrocytes upon activation of μ-opioid receptors, as it can be blocked by administration of the opioid receptor antagonist naltrexone [23,66]. This pro-inflammatory chemokine participates in the modulation of pain in response to opioids, reducing the analgesic property of morphine [67]. Additionally, the ability of morphine to induce glial proliferation and activation is associated with the activation of the mitogen-activated protein kinase pathway, a process that is mediated by CCL-5 [68]. Thus, CCL-5 may modulate glial proliferation and activation, playing an important role in opioid abuse and dependence [23]. A previous report showed a significant increase in the levels of the pro-inflammatory cytokines IL1β, TNF-α, and IL-6 in the neocortex and neostriatum of morphine addicted rats, though only when animals were undergoing spontaneous morphine withdrawal, while CCL-5 was significantly increased during the chronic consumption stage [66].

As has been reported for other drugs of abuse [69], morphine exposure can contribute to the neurobiology of addiction by altering glutamate and dopamine signaling via pro-inflammatory and pro-oxidative mechanisms [70,71,72]. In brain slice cultures, TNF-α has been shown to reduce glutamate transport, thereby increasing extracellular glutamate levels, which can lead to a hyperexcitable state [73]. Increases in extracellular glutamate levels have also been reported after an increase in brain oxidative stress [74,75]. Astrocytes surrounding the synapses are responsible for most of the glutamate reuptake from the synaptic clefts mediated by the action of the glutamate transporter GLT-1, playing a pivotal role in the termination of glutamatergic signal transmission and protecting neurons from the excitotoxic action of glutamate [76]. It has been reported that both neuroinflammation and brain oxidative stress are able to lower astrocyte GLT-1 levels [77]; additionally, morphine can directly induce the reduction of this glutamate transporter [78,79], leading to an increase in extracellular glutamate levels. Accordingly, we observed that 96 days of voluntary morphine intake in UChB rats led to a significant reduction in GLT-1 protein levels in the prefrontal cortex without altering the protein levels of either the cystine-glutamate antiporter xCT or the dopamine transporter DAT. The reduction of GLT-1 levels in these animals could lead to impairment of glutamatergic homeostasis, potentiating opioid dependence.

Polydrug use is a common condition among addicted patients [80], suggesting that a shared genetic vulnerability underlies different addictions. The results obtained with UChB rats, which self-administer ethanol, nicotine, and now opioids, support this concept and are in line with previous reports showing that the alcohol-naïve offspring of alcohol-preferring (P) rats selectively bred for high alcohol intake self-administered high amounts of nicotine compared to non-alcohol-preferring (NP) rats [81]. The increased drug self-administration of rats sensitive to polydrug use could be caused by a higher release of dopamine in the nucleus accumbens after drug exposure, as UChB rats release a higher amount of dopamine after the administration of 1 g/kg ethanol compared to rats selected for their alcohol avoidance [82]. Similar results have been reported for C57BL/6 mice, an inbred mouse strain that has high preference for alcohol, morphine, and cocaine, showing that C57BL/6 animals self-administered significantly more nicotine compared to the DBA/2 strain, which completely avoids alcohol [83,84]. Akin to UChB rats, C56BL/6 mice show an increased release of dopamine in the nucleus accumbens after direct drug administration, in this case 3 g/kg morphine, as compared to DBA/2 mice [85]. In the same line, cocaine and morphine both promoted significant conditioned placed preference, a marker of the rewarding effects of a drug, at lower dose in C57BL/6 than in DBA/2 mice [86], suggesting that shared factors mediate the co-abuse of different addictive drugs. 

One limitation of this study is the failure to address possible alterations of the dopaminergic system in the morphine-dependent animal model. It is noteworthy that while drugs of abuse acutely activate the dopaminergic system in naïve animals, human imaging studies have shown a reduction of dopamine receptors in the drug-dependent state, accompanied by a lesser degree of dopamine release in the ventral striatum of both heroin- and alcohol-dependent subjects, suggesting the existence of a “dopamine-impoverished” addicted brain. In this regard, the reader is referred to the review by Diana et al. [87]. Another limitation is that only female animals were used in this study. Thus, the extent of brain alterations induced by chronic voluntary morphine intake in male UChB rats is not known. 

## 4. Materials and Methods

### 4.1. Animals

Eight-week-old female naïve rats weighing 130–140 g of the UChB line selectively bred for their high voluntary oral ethanol intake [10,37] were single-housed at a constant temperature on a 12 h light/dark cycle (lights off at 7:00 p.m.) with unrestricted access to a soy protein and peanut meal rodent diet (Cisternas, Santiago, Chile) and water. All experiments were performed in female rats, as oral morphine consumption and its intravenous self-administration is higher in female than in male rats [88,89]. Additionally, females develop stronger dependence than males in animal models of multi-drug reward [90]. Animal procedures were approved by the Committee for Experiments with Laboratory Animals of the University of Chile (Protocol-CBA# 0994-FMUCH).

### 4.2. Drugs

Morphine hydrochloride trihydrate (20 mg/mL, Sanderson Laboratory, Santiago, Chile) was used for the initial intraperitoneal administrations and was injected at 40 mg/kg/day [36,91]. This morphine solution is formulated for intravenous or intraperitoneal administration. Morphine sulfate (20 mg/mL, Oramorph, Molteni Farmaceutici, Italy) was used to prepare the morphine solutions for oral consumption. Morphine sulfate solution concentrations for oral intake, calculated as morphine sulfate, ranged from 6 to 90 mg/L (*w*/*v*) and were prepared every day by dissolving morphine sulfate in distilled water. We opted for morphine sulfate for oral administration (Oramorph) because this solution is specifically formulated to enhance intestinal absorption following oral intake.

### 4.3. Induction of Voluntary Morphine Consumption

The voluntary oral consumption of morphine solution was induced as previously described [36]. Briefly, 50–60-day-old female UChB rats (n = 12) weighing from 130 to 140 g housed in individual cages were administered a dose of 40 mg/kg i.p. of morphine hydrochloride trihydrate daily for 9 consecutive days [36,91]. The intraperitoneal administration of morphine is essential to promote subsequent voluntary oral morphine intake. After discontinuation of morphine injections on day 10, each cage was fitted with a second drinking tube (a first water tube available all the time) containing a solution of morphine sulfate, with increasing concentrations on successive days. On day 10, the concentration of morphine sulfate in the second tube remained at 6 mg/L for three days. Subsequently, this concentration was incrementally increased from 5 to 10 mg every three or four days until it reached a maximum of 90 mg/L on day 37 to 40. On day 41 and onwards, rats were concurrently provided free-choice access to three bottles: one with water, and two others with different concentrations of morphine sulfate (80 and 90 mg/L) that were kept constant for 55 days. On day 42, the animals reached an average steady state of morphine consumption of 15.3 ± 0.22 mg/kg/day (mean ± SEM) and exhibited a greater preference for drinking morphine (83 ± 0.36%) over water. Oral voluntary morphine sulfate intake was expressed as mg/kg/day, while water intake was expressed as mL/kg/day. The above treatment conditions are depicted in Appendix A.

### 4.4. Naloxone-Precipitated Withdrawal

On day 50 of morphine intake, the same animals described above that had been voluntarily drinking 15.6 ± 0.8 mg/kg/day of morphine were intraperitoneally injected with a single dose (5 mg/kg, i.p.) of naloxone hydrochloride (Sigma-Aldrich, MO, USA) in a volume of 5 mL/kg. Immediately after naloxone administration, the animals were placed in a glass beaker (30 cm height and 18 cm diameter) and monitored for weight loss, diarrhea, forepaw tremors, chewing, jumping, climbing, and rearing events over 30 min as previously reported [61]. A withdrawal score was determined using an opioid deprivation rating scale obtained from Gellert et al. [92] with minor modifications. The scale weighed signs considering their frequency (graded signs) or valued their presence if minimum events were observed (checked signs). Graded signs include climbing (score 1 if frequency between 1 and 25, score 2 if between 25 and 50, score 3 if >50), jumping (score 1 if frequency between 1 and 5, score 2 if between 5 and 10, score 3 if >10), chewing (score 1 if frequency between 1 and 5, score 2 if between 5 and 10, score 3 if >10) and abdominal constrictions (score 1, every 2 events). Checked signs include wet-dog shakes (score 2 if >2), forepaw tremors (score 2 if >2), irritability (score 3 if observed), and diarrhea (score 3 if observed). As a control, the same dose of naloxone was intraperitoneally injected to animals drinking only water. After withdrawal precipitation, the animals were returned to their cages and continued voluntarily drinking morphine under the three-bottle choice paradigm containing water and 80 and 90 mg/L morphine solutions until day 96, reaching a consumption of 14.64 ± 0.52 mg morphine/kg/day (mean ± SEM). Additionally, a control group that had been initially injected (i.p.) with saline for 9 consecutive days (day -9 to day 0; see Appendix A) was used as a control for morphine injections and drank only water for 96 consecutive days. Immediately after completing 96 days of morphine or water intake, the animals were anaesthetized with a cocktail consisting of 60 mg/kg ketamine HCl, 10 mg/kg xylazine, and 4 mg/kg acepromazine administered intramuscularly in a volume of 1.9 mL/kg [93]. The animals were perfused intracardially with 100 mL of 0.1 M PBS (pH 7.4), then the brain was dissected and specific tissues were collected (snap frozen for protein and RNA determinations and fixed in 4% paraformaldehyde for immunofluorescence determinations).

### 4.5. Determination of Neurofilament-L (NF), Myelin Basic Protein (MBP), Myelin Proteolipid Protein (PLP), Glutamate Transporter 1 (GLT-1), Glutamate-Cystine Exchanger (xCT), and Dopamine Transporter (DAT) Levels in Prefrontal Cortex

Proteins from prefrontal cortex samples were extracted using T-per lysis buffer (Thermo-Fisher, MA, USA) containing protease inhibitors. For Western blots, 25 μg of proteins were utilized to assess NF levels with a rabbit anti-NF-L primary antibody (cat 2837, Cell Signaling, MA, USA 1:1000 dilution) and an IRDye 800CW goat anti-rabbit secondary antibody (cat 925-32211, Li-COR, NE, USA 1:10,000 dilution). MBP levels were examined using a chicken anti-MBP primary antibody (cat MBP0020, Aveslab, CA, USA, 1:1000 dilution) and an IRDye 680CW donkey anti-chicken secondary antibody (cat 926-68075, Li-COR, Li-COR, NE, USA 1:10,000 dilution). PLP levels were detected with a rat anti-myelin polypeptide protein primary antibody (cat MABN2620, Millipore, CA, USA, 1:1000 dilution) and an IRDye 680CW goat anti-rat secondary antibody (cat 925-68029, Li-COR, NE, USA, 1:10,000 dilution). The glutamate transporter GLT-1 was assessed using a guinea pig anti-GLT1 primary antibody (Cat AB1783, Millipore, CA, USA, 1:500 dilution) and an IRDye 800CW donkey anti-guinea pig secondary antibody (Cat 925-32411, Li-COR, NE, USA, 1:10,000 dilution). xCT levels were detected with a rabbit anti-xCT primary antibody (Cat AB175186, Abcam, Cambridge, United Kingdom, 1:500 dilution) and an IRDye 800CW donkey anti-rabbit secondary antibody (Cat 926-32213, Li-COR, NE, USA, 1:10,000 dilution). Finally, DAT levels were assessed using a rabbit anti-DAT primary antibody (Cat AB184451, Abcam, Cambridge, United Kingdom, 1:500 dilution) and an IRDye 800CW donkey anti-rabbit secondary antibody (Cat 926-32210, Li-COR, NE, USA, 1:10,000 dilution). The same membranes were probed for β-actin reactivity as a loading control, employing a mouse anti-β-actin primary antibody (Cat sc-47778, Santa Cruz Biotechnology, TX, USA, 1:200 dilution) and an IRDye 800CW goat anti-mouse secondary antibody (Cat 926-32210, Li-COR, NE, USA). Detection and quantification of reactive bands were performed using the Odyssey Imaging System (Li-COR) and analyzed with Image Studio Lite 5.2 software.

### 4.6. Determination of MBP and NF Immunoreactivity in Prefrontal Cortex

Double-labeling immunofluorescence against the mature oligodendrocyte marker MBP (cat MBP0020, Aveslab, CA, USA, 1:300 dilution) and the axonal marker NF (cat 2837, Cell Signaling, MA, USA, 1:300 dilution) was evaluated in coronal cryosections (30 μm thick) of the prefrontal cortex. Nuclei were counterstained with DAPI. Microphotographs were taken using a confocal microscope (Olympus FV10i). The area analyzed for each stack was 0.04 mm^2^, and the thickness (Z axis) was measured for each case. The mean fluorescence intensity of MBP and NF-L per mm^3^ was assessed using FIJI analysis software v1.8 as previously reported [94].

### 4.7. Determination of Astrocyte and Microglial Immunoreactivity in Nucleus Accumbens

Coronal cryosections (30 μm thick) of the nucleus accumbens underwent double-labeling immunofluorescence using the astrocyte marker glial fibrillary acidic protein (GFAP) (Sigma-Aldrich, MO, USA, G3893; 1:500 dilution) and the microglial marker ionized-calcium-binding adaptor molecule 1 (Iba-1) (Wako, VA, USA, 019-19741, 1:400 dilution). DAPI was employed for nuclear counterstaining. Confocal microscopy (Olympus FV10i) was utilized for capturing microphotographs. Each stack’s analyzed area was 0.04 mm^2^, and the *Z*-axis thickness was measured for each case. The density of GFAP-positive astrocytes and IBA-1-positive microglial cells was quantified using FIJI analysis software, as described previously [95].

### 4.8. Determination of mRNA Levels of Pro-Inflammatory Molecules in Nucleus Accumbens

Total RNA was extracted from the nucleus accumbens using TRIzol reagent (Invitrogen, Waltham, MA, USA) according to the manufacturer’s instructions. For reverse transcription, one microgram of total RNA was utilized with MMLV reverse transcriptase (Invitrogen) and oligo dT primers. Real-time PCR reactions were conducted in a final volume of 10 μL, including 50 ng cDNA, PCR LightCycler-DNA Master SYBRGreen reaction mix (Roche, IN, USA), 3 mM MgCl_2_, and 0.5 mM of the primers for the amplification of the pro-inflammatory molecules TNF-α, IL6, IL1β, and CCL-5 using a Light-Cycler 1.5 thermocycler (Roche, IN, USA), as previously reported [61]. To confirm mRNA origin and exclude genomic DNA interference, control reactions without reverse transcription (RT) were included. Expression levels were determined using the ΔΔCt method while normalizing the mRNA level of each target gene against the housekeeping gene glyceraldehyde 3-phosphate dehydrogenase (GAPDH) in the same sample.

### 4.9. Evaluation of Oxidative Stress in Neostriatum

Brain oxidative stress was evaluated in the neostriatum. The levels of lipid peroxidation were determining by assessing the amount of malondialdehyde (MDA) formed using the Lipid Peroxidation Assay kit (Sigma-Aldrich, MO, USA), as previously reported [36].

### 4.10. Statistical Analyses

The data are presented as means ± SEM. Statistical analyses were conducted using GraphPad Prism v.9.2.0 software. The normal distribution of data for all experiments was assessed using the Shapiro–Wilk test. For normally distributed data, statistical significance was determined using Student’s *t*-test. A level of *p* < 0.05 was considered significant. To facilitate the flow of the main text, full statistical analyses of each figure are presented in the legends of the respective figures.

## 5. Conclusions

Overall, present studies support the view that animals bred to prefer alcohol, which have been shown to self-administer nicotine, can develop a preference for opioids as well, suggesting the presence of a general drug-permissive genetic mechanism. in this study, these animals voluntarily ingested morphine solutions in amounts that led to clear morphine dependence, as shown by a marked naloxone-induced withdrawal reaction. Additionally, they exhibited the molecular and structural alterations commonly associated with loss of control over drug intake in humans chronically exposed to opioids, including cortical demyelination and axonal degeneration associated with striatal neuroinflammation, brain oxidative stress, and reduced GLT-1 levels. Thus, the present animal model is valuable for evaluating the broad spectrum of drug dependence-associated brain alterations induced by chronic morphine intake, and can help in assessing new therapeutic approaches for this devastating condition.

## Figures and Tables

**Figure 1 ijms-24-17081-f001:**
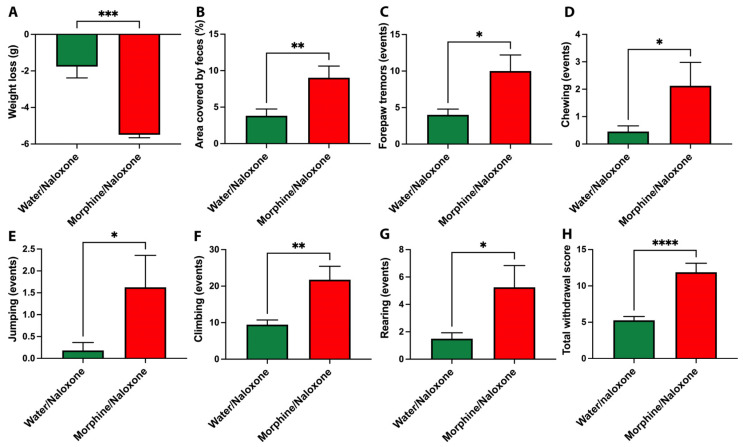
Naloxone administration induces a severe opioid deprivation syndrome in UChB rats that had voluntarily consumed morphine. Figure shows the weight loss (**A**), area covered by feces (%) (**B**), forepaw tremors (**C**), chewing (**D**), jumping (**E**), climbing (**F**), rearing (**G**), and total withdrawal score (**H**) of rats that had voluntarily consumed a morphine solution for a period of 50 days, the underwent intraperitoneal injection of 5 mg/kg naloxone, an opioid receptor antagonist. Control animals that consumed only water are included for comparison. Observations were made for 30 min post-naloxone administration and are presented as mean ± SEM. Statistical significance between the Water/Naloxone and Morphine/Naloxone groups was determined using a two-tailed Student *t*-test. * *p* < 0.05; ** *p* < 0.01 *** *p* < 0.001; **** *p* < 0.0001. n = 8 in each experimental group.

**Figure 2 ijms-24-17081-f002:**
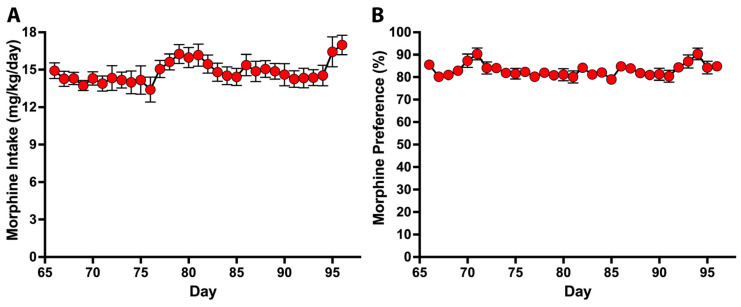
UChB rats exposed to a three-bottle choice voluntarily drink high amounts of morphine solution and show clear preference for ingesting the morphine solution over water. After confirming morphine dependence on day 50, animals continued to voluntarily drink morphine using the three-bottle choice paradigm (80 mg/L morphine sulfate, 90 mg/L morphine sulfate and water) until day 96. (**A**) Daily voluntary morphine intake (mg/kg/day) and (**B**) daily morphine preference over water. Data are shown as mean ± SEM. n = 6.

**Figure 3 ijms-24-17081-f003:**
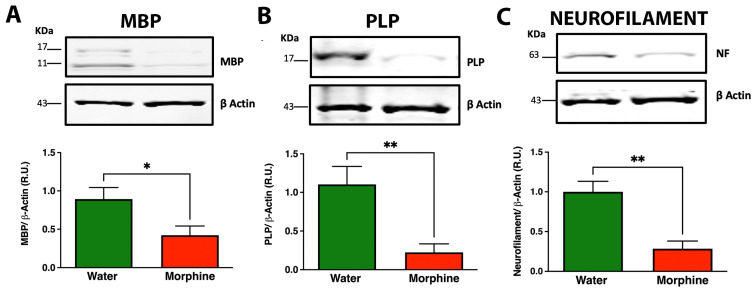
Chronic voluntary morphine intake leads to a significant reduction in myelin basic protein (MBP), myelin proteolipid protein (PLP), and neurofilament-L (NF) in the prefrontal cortex, indicating axonal demyelination. Western blot analysis of (**A**) MBP, (**B**) PLP, and (**C**) neurofilament-L in prefrontal cortex lysates of rats voluntarily drinking morphine for 96 days. Animals drinking only water were used as control. Data are presented as percentage ratios of MBP to β-actin, PLP to β-actin, and neurofilament-L to β-actin in the corresponding samples relative to control levels. Data are shown as mean ± SEM. Immunoblots shown are representative of n = 6 per experimental condition. Statistical significance between the water and morphine groups was determined using a two-tailed Student *t*-test * *p* = 0.035 for MBP, ** *p* = 0.0067 for PLP, ** *p* = 0.014 for Neurofilament.

**Figure 4 ijms-24-17081-f004:**
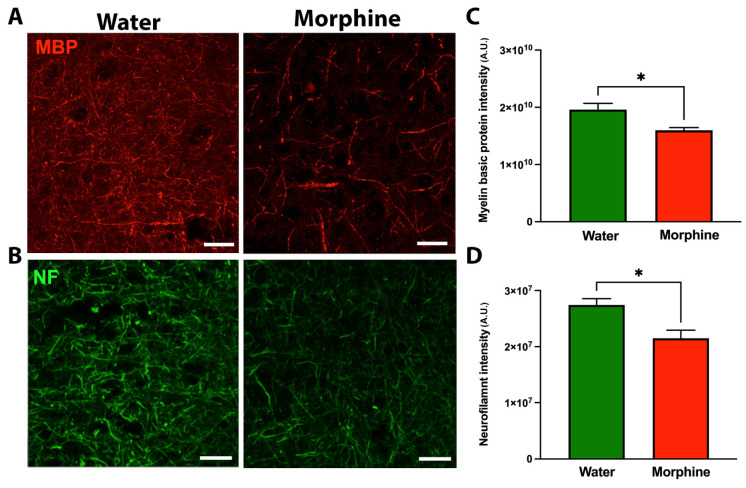
Chronic voluntary morphine intake reduces myelin basic protein (MBP) and neurofilament-L (NF) levels in the prefrontal cortex. Representative microphotographs of immunofluorescence against myelin basic protein (**A**) and neurofilament-L (**B**) in cryosections of the prefrontal cortex of rats voluntarily drinking morphine for 96 days. Animals drinking only water were used as control. Scale bar 25 μm. Quantification of myelin basic protein mean fluorescence intensity per mm^3^ expressed in arbitrary units (A.U.) (**C**) and neurofilament-L mean fluorescence intensity per mm^3^ expressed in arbitrary units (A.U.) (**D**). Data are shown as mean ± SEM. Immunofluorescence shown as representative of n = 6 per experimental condition. Statistical significance between the water and morphine groups was determined using a two-tailed Student *t*-test * *p* = 0.0109 for MBP, * *p* = 0.0053 for Neurofilament-L.

**Figure 5 ijms-24-17081-f005:**
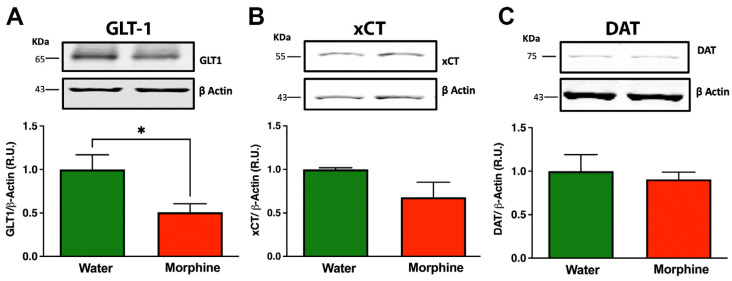
Chronic morphine intake reduces the levels of the glutamate transporter GLT-1 in the prefrontal cortex. Western blot analysis of the glutamate transporters GLT-1 (**A**) and xCT (**B**) and the dopamine transporter DAT (**C**) in prefrontal cortex lysates of rats drinking morphine voluntarily for 96 days. Animals drinking only water were used as control. Data are presented as percentage ratios of GLT-1 to β-actin, xCT to β-actin, and DAT to β-actin in the corresponding samples relative to control levels. Data are shown as mean ± SEM. Immunoblots shown are representative of n = 6 per experimental condition. Statistical significance between the water and morphine groups was determined using a two-tailed Student *t*-test * *p* = 0.031.

**Figure 6 ijms-24-17081-f006:**
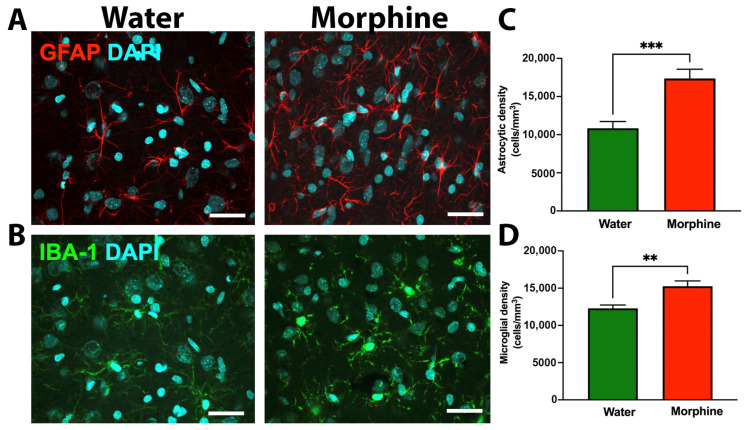
Chronic voluntary morphine intake increases astrocyte and microglial density in the nucleus accumbens. Representative immunofluorescence against GFAP (astrocytic marker) (**A**) and IBA-1 (microglial marker) (**B**) in cryosections of the nucleus accumbens of rats voluntarily drinking morphine for 96 days. Animals drinking only water were used as control. Scale bar 25 μm. Quantification of astrocyte density (GFAP^+^ cells/mm^3^) (**C**) and microglial density (IBA1^+^ cells/mm^3^) (**D**). Data are shown as mean ± SEM. Immunofluorescences shown are representative of n = 6 per experimental condition. Statistical significance between the water and morphine groups was determined using a two-tailed Student *t*-test ** *p* = 0.005, *** *p* = 0.0004.

**Figure 7 ijms-24-17081-f007:**
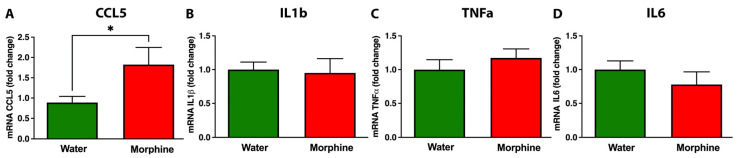
Chronic voluntary morphine intake increases CCL-5 mRNA levels in the nucleus accumbens. The mRNA levels of CCL-5 (**A**), IL1β (**B**) TNF-α (**C**), and IL6 (**D**) were determined by RT-qPCR in the nucleus accumbens of rats voluntarily drinking morphine for 96 days. Animals drinking only water were used as control. Data were normalized against the mRNA levels of the housekeeping gene GAPDH. Data are expressed as mean ± SEM of six animals per experimental condition. Statistical significance between the water and morphine groups was determined using a two-tailed Student *t*-test * *p* = 0.038.

**Figure 8 ijms-24-17081-f008:**
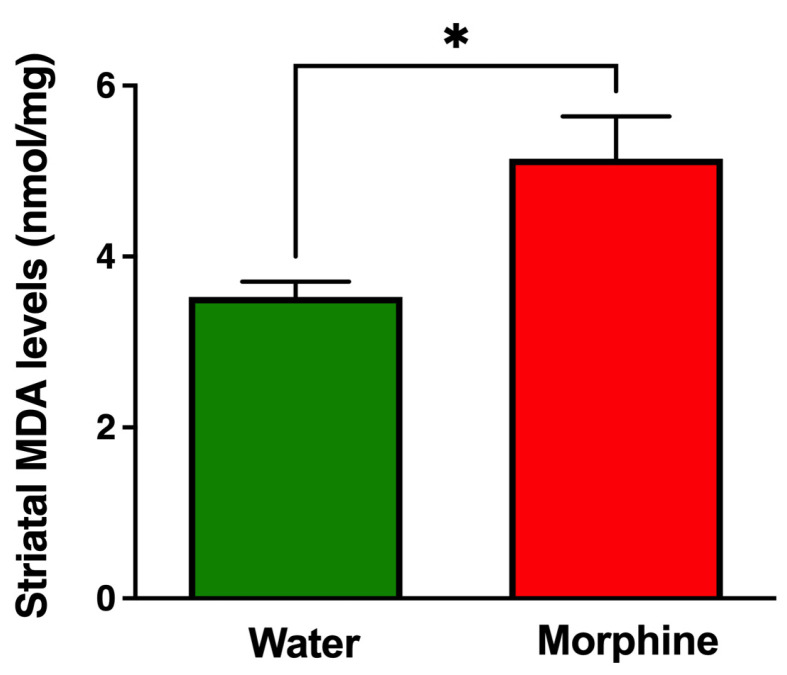
Chronic voluntary morphine intake increases malondialdehyde (MDA) levels in the neostriatum. The MDA levels were determined in the neostriatum of rats voluntarily drinking morphine for 96 days. Animals drinking only water were used as control. Data were normalized against mg of total proteins. Data are expressed as mean ± SEM of six animals per experimental condition. Statistical significance between the water and morphine groups was determined using a two-tailed Student *t*-test * *p* = 0.013.

## Data Availability

All data supporting this study are included within the article and Appendix A.

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
