# Peer review of "Chronic Voluntary Morphine Intake Is Associated with Changes in Brain Structures Involved in Drug Dependence in a Rat Model of Polydrug Use"

_ijms, 2023, doi:10.3390/ijms242317081_

Round 1
Reviewer 1 Report
Comments and Suggestions for Authors
After carefully reading the manuscript in my opinion, the Authors should focus their research on specific brain structures either involved in addiction or learning and memory. If the research is conducted mainly for memory and learning, this information should be included in the title.
There are stylistic and punctuation errors that need to be corrected and the font should be standardized.
In line 63 please add studies other than those conducted by the Authors.
Line 68 should include references typical for morphine results. Additionally, the Authors should specify exactly what types of memory they are writing about. Line 71-72 the mentioned structures are involved in rewarding activities rather than learning. This should be written and referred to in the title and aim of the manuscript.
Line 93 why did the Authors initially administer morphine to rats ip?
Last paragraph of introduction should be extended by information about morphine influence on level of other neurotransmitters than glutamate.
Studies involving only female rats may be considered unreliable when there is no comparison to males. Most current studies concern males in order to eliminate, among others, influence of the monthly cycle on animal behavior. The Authors should comment on whether similar studies have been conducted for males. If so, what results were obtained?
Why did the Authors administer a different morphine compound for drinking and another for injection?
Line 394 priming dose is an inappropriate term here. This type of term is used when we are dealing with the relapse phenomenon and not daily administration.
The Authors should provide exact one-way ANOVA values.
In section results Authors included description of methodology. It should be deleted, e.g. in point 2.1 lines from 106 to 118 should be deleted. Similarly, in other sections describing the results, only the results should be left.
In their manuscript, the Authors draw attention to the influence of morphine on learning several times, but nowhere is this issue discussed in detail. If they want to focus on this issue, it is necessary to rewrite the manuscript.
The aim of the study is not clearly formulated - it is difficult to say whether it is about the effect of morphine on learning, multi-drug addiction or perhaps neurotransmission. Additionally, as the Authors themselves emphasize, the presented data have often already been published, so it is necessary to justify what is the novelty of research.
Lines 354-372 should rather be moved to introduction.
The discussion should be re-read and re-edited. In my opinion, in its current form it is only an extended description of the results and an emphasis that other authors have obtained similar results. After all, it should be about collecting information and drawing conclusions.
Conclusions also do not add anything new to the work and the first sentence: „Overall, present studies support the view that animals breed to prefer one type of drug can also develop a preference for other drug types, suggesting the presence of general drug-permissive genetic mechanism”. Basically, it is a summary of research that has been known for years and only intensifies the question about novelty in the Authors' research.
There are no limitations of the study. They should be added.
Author Response
Reviewer #1
Rev#1-Comment 1: After carefully reading the manuscript in my opinion, the Authors should focus their research on specific brain structures either involved in addiction or learning and memory. If the research is conducted mainly for memory and learning, this information should be included in the title.
Rev#1 Reply 1: Thank you for this comment. We agree with the reviewer’s suggestion that the manuscript title should be primarily focused. We have now rewritten the title and the manuscript, focusing on brain structures involved in drug dependence rather than addressing learning and memory.
Rev#1 Modification 1: Following Editor’s suggestion we changed the manuscript title to “Chronic voluntary morphine intake is associated with changes in brain structures involved in drug dependence in a rat model of polydrug use”.
Rev#1-Comment 2: There are stylistic and punctuation errors that need to be corrected and the font should be standardized
Rev#1 Reply 2: Thank you very much for bringing these errors to our attention.
Rev#1 Modification 2: We have carefully reviewed the entire document and corrected all detected errors.
Rev#1-Comment 3: In line 63 please add studies other than those conducted by the Authors.
Rev#1 Reply 3: Many thanks for this comment.
Rev#1 Modification 3: In response to the reviewer's suggestion, we have included additional studies (references 14 and 15) demonstrating that various drugs effective in reducing alcohol intake and relapse have also shown efficacy in decreasing nicotine intake.
Rev#1-Comment 4: Line 68 should include references typical for morphine results. Additionally, the Authors should specify exactly what types of memory they are writing about. Line 71-72 the mentioned structures are involved in rewarding activities rather than learning. This should be written and referred to in the title and aim of the manuscript.
Rev#1 Reply 4: Many thanks for this comment.
Rev#1 Modification 4: Addressing the reviewer's suggestion, we have included additional references for the results related to morphine (references 17 and 18). Additionally, we specifically reference the prefrontal cortex and nucleus accumbens as being primarily associated with drug dependence and loss of control over drug intake. Therefore, the work addressing memory per se was removed.
Rev#1-Comment 5: Line 93 why did the Authors initially administer morphine to rats ip?
Rev#1 Reply 5: Thank you for your comment. Indeed, the rationale behind the intraperitoneal administration of morphine was not clear. Rats exhibit a natural aversion to consuming morphine solutions due to its bitter taste. Therefore, our initial step was to administer morphine intraperitoneally to induce a level of dependence and thus encourage voluntary consumption of morphine.
Rev#1 Modification 5: In Introduction section we added the following paragraph “In recent studies, we showed that UChB rats, following one-week intraperitoneal morphine administration, voluntarily consumed escalating amounts of morphine until reaching a steady state [36]. Rats typically avoid consuming morphine solutions due to their bitter taste. Therefore, intraperitoneal morphine administration enables the induction of a level of dependence, subsequently promoting voluntary morphine consumption [36].
Rev#1-Comment 6: Last paragraph of introduction should be extended by information about morphine influence on level of other neurotransmitters than glutamate.
Rev#1 Reply 6: We very much appreciate this recommendation, we have now indicted the effects on dopamine tone, but rather inserted in the Discussion section.
Rev#1 Modification 6: In Discussion section we added the following paragraph: “It is noteworthy that while in naïve animals drugs of abuse acutely activate the dopaminergic system, in the drug-dependent state, human imaging studies have shown a reduction of dopamine receptors accompanied by a lesser degree of dopamine release in the ventral striatum of both heroin and alcohol-dependent subjects, thus suggesting the existence of a “dopamine-impoverished” addicted brain. In this regard, the reader is referred to the review by Diana et al [87].
Rev#1-Comment 7: Studies involving only female rats may be considered unreliable when there is no comparison to males. Most current studies concern males in order to eliminate, among others, influence of the monthly cycle on animal behavior. The Authors should comment on whether similar studies have been conducted for males. If so, what results were obtained?
Rev#1 Reply 7: Many thanks for this comment. The main reason for the use of female rats is that oral morphine consumption and its intravenous self-administration is higher in female than in male rats (Cicero, T.J et al. Gender differences in the intravenous self-administration of mu opiate agonists. Pharmacol Biochem Behav 2003, 74, 541-549). Additionally, in animal models of multi-drug reward females develop stronger dependence than males (Carroll ME, et al. Intravenous cocaine and heroin self-administration in rats selectively bred for differential saccharin intake: phenotype and sex differences Psychopharmacology 2002, 161:304–313). However, studies have been conducted for male animals, showing similar structural and molecular brain alterations (Boronat, M.A et al. Attenuation of tolerance to opioid-induced antinociception and protection against morphine-induced decrease of neurofilament proteins by idazoxan and other I2-imidazoline ligands. Br J Pharmacol 1998, 125, 175-185); (Campbell, L.A. et al. CCL5 and cytokine expression in the rat brain: differential modulation by chronic morphine and morphine withdrawal. Brain Behav Immun 2013, 34, 130-140).
Rev#1 Modification 7: In the Materials and Methods section, we incorporated a paragraph explaining the rationale behind the use of female animals.
Rev#1-Comment 8: Why did the Authors administer a different morphine compound for drinking and another for injection?
Rev#1 Reply 8: Thank you for this comment, which enables us to provide a clearer explanation of the rationale behind the selection of the morphine solutions used in this study. We opted for morphine sulfate for oral administration (Oramorph) because this solution is specifically formulated to enhance intestinal absorption following oral intake. In contrast, morphine chloride is formulated for intravenous or intraperitoneal administration.
Rev#1 Modification 8: In the Material and Methods section we have included this information.
Rev#1-Comment 9: Line 394 priming dose is an inappropriate term here. This type of term is used when we are dealing with the relapse phenomenon and not daily administration.
Rev#1 Reply 9: Many thanks for this comment.
Rev#1 Modification 9: The term priming was deleted.
Rev#1-Comment 10: The Authors should provide exact one-way ANOVA values.
Rev#1 Reply 10: We apologize for the mistake. Since we are comparing two experimental groups, after checking the normal distribution of data, statistical significance was determined using Student’s t-test.
Rev#1 Modification 10: In the Materials and Methods sections, we corrected the information regarding the statistical test used. Additionally, in each figure legend, we provided the exact p-value.
Rev#1-Comment 11: In section Results authors included description of methodology. It should be deleted, e.g. in point 2.1 lines from 106 to 118 should be deleted. Similarly, in other sections describing the results, only the results should be left.
Rev#1 Reply 11: Many thanks for this suggestion. However, we believe that those few methodology sentences before each presented result are crucial for understanding the rationale behind each of the conducted experiments and could assist the reader in better comprehending the work. Therefore, if the reviewer allows, we would prefer not to remove these sentences from the Results section.
Rev#1 Modification 11: No modification was implemented.
Rev#1-Comment 12: In their manuscript, the Authors draw attention to the influence of morphine on learning several times, but nowhere is this issue discussed in detail. If they want to focus on this issue, it is necessary to rewrite the manuscript.
Rev#1 Reply 12: Many thanks for this important comment. We completely agree. Thus, we decided to focus our manuscript on drug dependence rather than learning and memory.
Rev#1 Modification 12: Following the reviewer's suggestion, we narrowed the focus of the manuscript towards the structural and functional alterations associated with drug dependence and loss of control over drug intake.
Rev#1-Comment 13: The aim of the study is not clearly formulated - it is difficult to say whether it is about the effect of morphine on learning, multi-drug addiction or perhaps neurotransmission. Additionally, as the Authors themselves emphasize, the presented data have often already been published, so it is necessary to justify what is the novelty of research.
Rev#1 Reply 13: Many thanks, the main aim of our study is now centered on dependence, since prefrontal dysfunction (recall and memory) also leads to loss of control over drug use; thus, indeed affecting dependence.
Rev#1 Modification 13: The afore mentioned modifications were incorporated throughout the manuscript and the title was changed to reflect this aim.
Rev#1-Comment 14: Lines 354-372 should rather be moved to introduction.
Rev#1 Reply 14: Many thanks for this comment. However, since we are comparing the results obtained in the UChB rats with those reported for other animal models of polydrug use, we think that the best place for this information is in the Discussion section.
Rev#1 Modification 14: Hoping that the reviewer will allow us to reduce the length of the Introduction, we have left this paragraph in the Discussion section. No modification was implemented.
Rev#1-Comment 15: The discussion should be re-read and re-edited. In my opinion, in its current form it is only an extended description of the results and an emphasis that other authors have obtained similar results. After all, it should be about collecting information and drawing conclusions.
Rev#1 Reply 15: Many thanks for this comment.
Rev#1 Modification 15: Following reviewer’s suggestion the discussion was completely rewritten.
Rev#1-Comment 16: Conclusions also do not add anything new to the work and the first sentence: Overall, present studies support the view that animals breed to prefer one type of drug can also develop a preference for other drug types, suggesting the presence of general drug-permissive genetic mechanism”. Basically, it is a summary of research that has been known for years and only intensifies the question about novelty in the Authors' research.
Rev#1 Reply 16: Many thanks, we agree that such may be the perception. However, not all animal lines bred to prefer one drug type engage in consuming another drug of abuse. It has been reported that rats bred to prefer an opioid do not show a preference for ethanol or cocaine, suggesting a directional susceptibility (45). However, we agree that the Conclusion required mayor changes that were implemented.
Rev#1 Modification 16: Following reviewer’s suggestion Conclusion was changed as follows “Overall, present studies support the view that animals bred to prefer alcohol, which also self-administer nicotine, can also develop a preference for opioids, suggesting the presence of a general drug-permissive genetic mechanism. These animals voluntarily ingested morphine solutions in amounts that led to clear morphine dependence, as shown by a marked naloxone-induced withdrawal reaction. Additionally, they exhibited the molecular and structural alterations commonly associated with loss of control over drug intake in human chronically exposed to opioids, including cortical demyelination and axonal degeneration, associated with striatal neuroinflammation, brain oxidative stress, and reduction of GLT-1 levels. Thus, the present animal model is valuable for evaluating the broad spectrum of drug-dependence associated brain alterations induced by chronic morphine intake and for assessing new therapeutic approaches for this devastating condition”.
Rev#1-Comment 17: There are no limitations of the study. They should be added.
Rev#1 Reply 17: Many thanks for this comment.
Rev#1 Modification 17: Following reviewer’s suggestion, at the end of the Discussion section, we have highlighted limitations of the study. These included not exploring dopamine as an additional neurotransmitter that could be affected by chronic morphine intake, and the fact that only one animal sex was studied.

Reviewer 2 Report
Comments and Suggestions for Authors
The study would be more interesting if the Authors provided another group of animals, i.e. rats pretreated with saline but with free access to either morphine or water.
Do all the brain changes in UChB rat differ from those observed in other lines of rats that are no bred for their voluntary ethanol intake? This should be taken into consideration while presenting discussion
The treatment period is quite long, therefore I was wondering whether the Authors have some of the results for weight loss (last day vs. first day), or any additional side effects?
Comments on the Quality of English Languageminor changes are required
Author Response
Reviewer #2
Rev#2-Comment 1: The study would be more interesting if the Authors provided another group of animals, i.e. rats pretreated with saline but with free access to either morphine or water.
Rev#2 Reply 1: Many thanks for this comment. Indeed, we included this group of animals in the original experimental design. However, without the preconditioning strategy (intraperitoneal morphine administration), animals do not drink a sufficient amount of morphine to generate dependence and were removed from the study. It is well known that the bitter taste of morphine causes animals to reject this solution and prefer to consume water, preventing the attainment of the necessary morphine concentrations to induce dependence. This can be avoided through intraperitoneal pretreatment with morphine, inducing initial drug dependence. Once discontinued, the intraperitoneal administration leads the animals to voluntarily consume increasing morphine concentration despite its bitter taste.
Rev#2 Modification 1: In Introduction section, Materials and Methods section and in Results section, we included a sentence indicating that the intraperitoneal administration of morphine is essential to promote subsequent voluntary oral morphine intake.
Rev#2-Comment 2: Do all the brain changes in UChB rat differ from those observed in other lines of rats that are no bred for their voluntary ethanol intake? This should be taken into consideration while presenting discussion
Rev#2 Reply 2: Many thanks for this question. As previously indicated in response to reviewer 1, not all animal lines bred to prefer one drug type engage in consuming another drug of abuse. It has been reported that rats bred to prefer an opioid do not show a preference for ethanol or cocaine (45), suggesting a directional susceptibility. As for specific brain changes, such study did not examine brain changes but dealt only with behavioral differences.
Rev#2 Modification 2: Following reviewer’s suggestion in Discussion section we added a paragraph presenting this information.
Rev#2-Comment 3: The treatment period is quite long, therefore I was wondering whether the Authors have some of the results for weight loss (last day vs. first day), or any additional side effects?
Rev#2 Reply 3: Many thanks for this important question. Indeed, we measured body weight every four days and total liquid intake daily as indicators of side effects. However, no alterations over time were observed in both parameters compared to control animals not exposed to morphine. It has been previously reported that when morphine consumption is progressively increased over time, as in the morphine access scheme utilized in the present study, no weight loss is induced (Martin R et al. Psychopharmacologia, 1963, 4, 247–260).
Rev#2 Modification 3: Following the reviewer's suggestions, we added the Supplementary Figure 3 to the manuscript, displaying the data on body weight during all the experiment. These data were also mentioned in Result section.
Round 2
Reviewer 1 Report
Comments and Suggestions for Authors
In my opinion, the manuscript has been sufficiently revised by the Authors.